# A CNT-Toughened Strategy for In-Situ Repair of Aircraft Composite Structures

**DOI:** 10.3390/ma15217691

**Published:** 2022-11-01

**Authors:** Tengfei Yang, Shiyong Chu, Bin Liu, Fei Xu, Bo Wang, Chengwei Wu

**Affiliations:** 1School of Aeronautics, Northwestern Polytechnical University, Xi’an 710029, China; 2Aviation Industry Development Research Center of China, Beijing 100028, China; 3State Key Laboratory of Structural Analysis for Industrial Equipment, Department of Engineering Mechanics, Dalian University of Technology, Dalian 116024, China

**Keywords:** CNT, in-situ repair, fracture toughness, cohesive-interface modeling

## Abstract

This study aimed to develop an in-situ field-repair approach, especially for aircraft composite structures, to enhance the interlaminar toughness of plain-woven composites (PWCs) by adding multi-walled carbon nanotubes (MWCNTs). MWCNTs were dispersed at each interface between prepreg layers by means of solvent spraying, with a density of 1.58 g/m^2^. Then, the layers were stacked with the predefined sequence and cured at 120 °C and 1 bar pressure, using a heat-repairing instrument. A standard double cantilever beam (DCB) test was used to investigate the interlaminar toughening effect that was due to the MWCNTs. For comparison, original samples were also prepared. The results indicated that the introduction of MWCNTs can favorably enhance the interlaminar toughness of PWCs in a field-repair approach and the Mode I fracture energy release rate, *G_IC_*, increased by 102.92%. Based on the finite element method (FEM) of continuum damage mechanics, the original samples and the MWCNTs toughening specimen under DCB Mode I fracture were modeled and analyzed. The simulation and the experiment were in good agreement. Finally, when the toughening mechanism of MWCNTs was explored with a scanning electron microscope (SEM), we found that a large amount of fiber-matrix (F-M) interface debonding and matrix cracking in mountain shape were the major modes of fracture, accompanied by a small amount of fiber breakage and matrix peeling for the MWCNTs-toughening specimens.

## 1. Introduction

Carbon fiber reinforced polymer (CFRP) composites are being increasingly used in aeronautical fields due to their excellent properties, such as high strength, high modulus, light weight, good heat resistance, and good fatigue resistance.

Composite materials account for an increasing proportion of the new generation of materials in advanced aircrafts, such as the Boeing 787 and the Airbus A350 XWB, in which composite materials account for more than fifty percent of the materials. Even though CFRP composites have excellent in-plane properties, they are susceptible to delamination caused by low-velocity impacts (e.g., dropped tools, hail, and sand), due to their low interlaminar toughness. Moreover, delamination is also likely to reduce the mechanical properties of CFRP composites, especially the compressive strength after impact, which is the key indicator of impact-damage tolerance [1,2]. Therefore, it is necessary to study the interlaminar fracture performance of CFRP composites, identify their mechanisms, and explore ways to improve them. 

In most previous studies, researchers have examined the interlaminar strength and fracture toughness of CFRP composites using different methods and various perspectives. Several techniques have been successfully devised to enhance delamination resistance [3,4,5,6,7,8], such as designing 3D fabric architecture [3,4], transversely stitching or pinning the fabrics [5,6], fiber hybridization [7,8], toughening the matrix material [8], and placing interleaves between the plies [8]. These technologies improved interlaminar fracture toughness, while sacrificing in-plane mechanical properties [9] or increasing the thickness of CFRP laminates. 

In 1991, carbon nanotubes (CNTs) [10] invented by Lijima provided a promising opportunity to significantly enhance interlaminar toughness without weakening in-plane properties or enlarging the thickness of CFRP laminates. Hamer et al. [11] utilized multi-walled carbon nanotubes (MWCNTs) reinforced by electrospun Nylon 66 interleaving to improve the Mode I and Mode II fracture toughness of CFRP laminates. MWCNTs-reinforcement can effectively resist cracks from continuously propagating between adjacent plies. The reinforced CFRP laminates required 400% and 160% Mode I and Mode II fracture energies, respectively, compared with the non-reinforced CFRP laminates. 

Joshi et al. [12] proposed a novel method for dispersing MWCNTs onto woven CFRP prepregs, and the interlaminar fracture toughness of CFRP laminates with different densities of MWCNTs were evaluated by double cantilever bending (DCB) and end-notched flexure (ENF) tests. The results showed that the addition of MWCNTs between the CFRP prepregs helped in strengthening the interface and fracture toughness, and the optimal MWCNTs density was identified as 1.32 g/m^2^ at the midplane, in which *G_IC_* and *G_IIC_* were enhanced by 32% and 140% at crack propagation, respectively. 

Tong et al. [13] proposed two continuum-mechanics-based models to identify the pull-out mechanisms of long MWCNTs by theoretical analysis and experimental observations. These two models were employed to investigate the influence of long MWCNTs on the Mode I delamination toughness of laminated composites by numerical simulations of DCB tests. The results indicated that the addition of MWCNTs improved the Mode I interlaminar fracture toughness of the composites, and the density, length, and maximum pull-out displacement of MWCNTs have significant effects on Mode I fracture toughness.

Khan and Kim [14] fabricated bucky papers using CNTs or carbon nanofibers (CNFs), which were subsequently integrated and cured with carbon fiber prepregs to produce CFRP laminates. The shear strength and flexural properties of the obtained CFRP laminates were explored by experiments. It was found that interlaminar shear strength and Mode II toughness were increased by 31% and 104%, respectively, for the multiscale CFRP composites. 

Almuhammadi et al. [15] dispersed MWCNTs on the interfaces between prepregs; the DCB experiments showed that the Mode I fracture toughness of the nano-reinforced composites was improved by 17%. Moreover, according to the scanning electron microscope (SEM) analyses of the damaged morphologies, the improvement was mainly attributed by the ability of nano-reinforcement to prevent the initiation and propagation of damage by the pullout, peeling, and bridging of CNTs. 

Liu et al. [16] reported a low-cost approach by adding two different nano-materials, MWCNTs and multi-layers graphene (mG), to the interlayer surface. The experimental DCB tests indicated that the values of *G_IC_* were enhanced by 12.3% and 101.4%. respectively, which was in good agreement with the simulation results. 

In summary, in these previous studies, researchers mostly focused on autoclave or autoclave-like processes of enhancing the interlaminar toughness of composites by adding nano-materials (CNTs and MWCNTs).

An autoclave is a special pressure vessel that can bear and regulate a certain temperature and pressure range. Autoclave molding refers to the process of using the high temperature and compressed gas inside an autoclave to generate pressure to heat and cure the composite prepregs. Accordingly, for the autoclave process, the main features are the mold that ensures the shape of the structure and the high pressure of more than 1 bar on the surface of the composite prepregs. These features make the autoclave process suitable for manufacturing or repairing the integral structure in a factory. However, in wartime or other emergencies, the damaged parts of composite structures must be repaired effectively and as soon as possible. In the autoclave process, repairing is complicated and time-consuming. Moreover, the damaged parts of the composite structures need to be disassembled, if possible, while large non-detachable components must be placed in an autoclave of a large or super-large size. This results in increasing the three-dimensional size of the autoclave, which is particularly unacceptable.

Because of the repair problems of the autoclave process, we focused on the field-repair process of composite structures. Unlike the autoclave process, the field-repair process is low-cost and efficient, and it requires no complex instruments or equipment. For different types of damage, the field-repair process requires no special mold; rather, it relies on the original composite structure, initially grinding in the damaged components and then making the repairs. Moreover, the field-repair process only requires 1 bar of curing pressure by a vacuum bag, which is lower than the high pressure required by the autoclave process. Although the field-repair process has several advantages in recovering damaged composite structures, the question remains whether the mechanical properties of the post-repaired structures can be kept consistent with those of undamaged structures. In addition, whether the improved process possesses significant influence for structural safety continues to bean important concern for academic and engineering researchers.

Some recent investigations have developed field-repair approaches and studied post-repaired performance. In the work of Karuppannan et al. [17], a series of field-repair works were carried out according to the damage situation, including repair patch design, fabrication, and implementation of the repair. That work foundd that CFRP patching is a feasible choice for repairing damaged metal structures. Monsalve et al. [18] selected CNTs as a reinforcement in epoxy resin to decrease the fatigue crack propagation rate in a 2024 T3 Al alloy. The results showed that a small amount of CNTs is helpful in extending the life of Al alloy structures with fatigue crack propagation. in The crack initiation and propagation were dramatically delayed. When the resin matrix included 0.5 vol% and 1 vol% CNTs reinforcement, the fatigue life increased by 104% and 128%, respectively. Kong et al. [19] simulated the impact damage of unidirectional CFRP composite spar structures and sandwich structures. Then, they repaired the damaged structures using the external patch repair method and compared the compressive strengths of the repaired structures with the undamaged ones. They found, experimentally and numerically, that the compressive strengths of the repaired spar and sandwich structures recovered to 91.19% and 88.68%, respectively, compared with the undamaged ones.

Due to the excellent improvement in the mechanical properties of CFRP composites by adding MWCNTs [16,20], this study developed a field-repair approach to enhance the interlaminar toughness of plain-woven composites (PWCs) by adding MWCNTs into each interface between prepreg layers. The field-repair strategy and the experimental work are presented in Section 2. The experimental and simulation results are discussed and analyzed in Section 3, and the conclusions are provided in Section 4.

## 2. Materials, Specimen Manufacturing, and Experimental Procedures

### 2.1. Materials

The fabrics contained carbon fiber T700-3K (diameter: 7 μm) provided by the Toray company, and the epoxy resin was provided by the Loctite company. Their material properties are listed in Table 1. The MWCNTs were manufactured by the chemical vapor deposition (CVD) approach, and the purity was greater than 95 wt%. The length of the MWCNTs varied from 3 μm to 12 μm, and the internal and external diameters were 3 nm to 5 nm and 8 nm to 15 nm, respectively. Polyethylene (PE) films with a thickness of 0.03 mm were inserted at the midplane of the laminate during layup to form an initiation site for the delamination. The total length was approximately 75 mm.

### 2.2. Specimen Manufacturing

In this study, MWCNTs were utilized to enhance the interlaminar toughness of PWC laminates by a low-cost and rapid field-repair strategy. The MWCNTs-based field-repair technological process for composite specimens is shown in Figure 1. The first step was dispersing the MWCNTs into the acetone solution under an ultrasonic wave for 15 min at 23 °C (ambient temperature) to avoid the agglomeration of MWCNTs, and spraying the nano-materials solution on each surface of the prepregs by a sprayer. The composite prepregs were made by coating the epoxy resin uniformly on the carbon-fiber cloth. The supplement’s density of 1.58 g/m^2^ was chosen to investigate and obtain the interlaminar toughness enhancement. Then, 20 wet composite prepregs were totally stacked with the predefined sequence, in which the 0° and 90° plies were alternately laid. Polyethylene (PE) films were inserted between the tenth and eleventh layers to represent the initial Mode I cracks (i.e., the LVI-induced delamination). After that, the wet fabrics were cured by a heat-repairing instrument with a curing temperature of 120 °C and a pressure of 1 bar for 4 h.

### 2.3. Set-up and Equipment

The heat-repairing instrument is a comprehensive repair tool for the composite structures that was capable of heating up to 760 °C and vacuuming. Due to its excellent convenience and reliability, this equipment is usually used to provide the curing temperature and the vacuum environment for the resin curing. The heating rate, temperature maintenance time, and cooling rate can be set according to the operation requirements. In addition, the heat-repairing instrument possesses significant advantages in repairing composite structures with large dimensions and complex shapes, because it is not required to disassemble the structural parts.

### 2.4. Double Cantilever Beam (DCB) Set-Up and Tests

CFRP laminates with a thickness of 7 mm were obtained. In addition, a group of original laminates were fabricated as a reference to evaluate the interlaminar toughening effect caused by MWCNTs. The Mode I fracture toughness tests were performed on the obtained DCB specimens according to the ASTM D5528-13 standard [21], as a reference. Figure 2 shows the schematic illustration of the DCB specimen and the loading conditions. A pair of piano hinges were bonded adhesively on the top and bottom sides of each DCB specimen, near to the inserted PE film. The piano hinges were placed with the fixtures of a universal tensile-testing machine. Tensile displacement perpendicular to the specimen’s length direction was carried out on the upper piano hinge to prompt crack growth. In addition, a quasi-static displacement rate of 1 mm/min was used during the loading process.

During the DCB testing process, the applied load and the opening displacement were recorded. The onset of the delamination (i.e., the Mode I crack growth) was defined by the sudden drop of the load-displacement curve. After the maximum load, the applied load nonlinearly decreased as the opening displacement continuously rose. The delamination length was measured to calculate the Mode I fracture toughness. To evaluate the enhancement of interlaminar toughness, a group of original DCB specimens were manufactured without adding MWCNTs. Furthermore, each testing group included four specimens to ensure the reproducibility of the experimental results. Details of the experimental results are presented in Section 3.1.

In order to eliminate the influence of the measured crack length on the fracture toughness, the total crack length *a* (Figure 3) can be calculated by Equation (2), and the Mode I energy release rate *G_IC_* was calculated by Equation (3) [22].
(1)$$I_{z}=\frac{1}{12}bh^{3}$$
(2)$$a=\sqrt[{3}]{\frac{3\delta E_{11}I_{Z}}{2P}}$$
where *I_z_* is the moment of inertia of one arm, *b* is the width of the specimen, *h* is half the thickness of the specimen, *P* is the applied load, *δ* is the applied displacement, and *E*_11_ is the elastic modulus of the composite materials.
(3)$$G_{IC}=\frac{P^{2}}{E_{11}I_{z}b}\sqrt[{\frac{3}{2}}]{\frac{3\delta E_{11}I_{z}}{2P}}$$

## 3. Experimental and Simulation Results and Analysis

### 3.1. Experimental Behavior, Crack Length a and Energy Release Rate G_IC_

The load-displacement curves of the original group (four specimens) and MWCNTs toughening group (four specimens) are depicted in Figure 4. As displayed in Figure 4, the load-displacement curves possessed two representative characteristics: (a) the MWCNTs toughening group had a larger opening displacement, and (b) the MWCNTs toughening group possessed the larger maximum peak value of the load (P_max_) and the load dropped more slowly in the descent stage. That descent stage is not a sudden process, so overall it is more smooth.

In order to intuitively characterize the process of the DCB tests of tensile, Figure 4 was simplified to Figure 5, which shows the three obvious stages: the linear increasing stage, the nonlinear increasing stage, and the nonlinear decreasing stage. As shown in Figure 5, after the linear increasing stage and nonlinear increasing stage, the load reached the maximum value and then entered the nonlinear decreasing stage. The maximum loads and nonlinear decreasing slopes of the original specimens and the MWCNTs toughening ones are listed in Table 2.

As shown in Table 2 with regard to the MWCNTs toughening specimens, the average value of P_max_ increased by 19.5% and the slope of nonlinear decrease stage decreased by 63.8%, compared with the original specimens.

The crack-growing length *a* and the energy-release rate *G_IC_* were calculated by Equation (1) and Equation (2), respectively, as shown in Figure 6. The initiation values of *G_IC_* were determined using the load and deflection measured at the point of deviation from linearity in the load-displacement curve, which is typically the lowest among the three different *G_IC_* initiation value definitions. However, this paper focused on a comparison of the fracture toughness of the original specimens and the MWCNTs toughening specimens, so that influence was ignored. The crack propagation was divided into two stages: the crack initiation non-stable stage (the blue rectangular box in Figure 6) and the crack propagation stable stage (the red rectangular box in Figure 6). It was reasonable to choose the data from the stable crack propagation stage as the *G_IC_* data source, as shown in Figure 6.

The calculated results, as shown in Figure 7, showed that the critical energy-release rate *G_IC_* of MWCNTs toughening specimens in the stable crack-propagation stage obtained improvement of 102.92%, from 1199.15 J/m^2^ (original specimens) to 2435 J/m^2^ (MWCNTs toughening specimens). Even when a group of highest value (MWCNTs-4) was removed, it still showed 95% improvement. The data for MWCNTs toughening specimens were imperfect and more highly dispersed than the data for the original specimens. The above analysis indicates that the interlaminar Mode I toughness of CFRPs can be obviously enhanced by embedding MWCNTs between the adjacent laminas with the heat-repairing instrument in the field-repair condition.

### 3.2. Simulation Method and Results

#### 3.2.1. Cohesive Damage Theory

A built-in cohesive element with a bilinear traction-separation relationship in ABAQUS/Standard was utilized to capture the delamination. The elastic response of the cohesive element was governed as follows:(4)$$\left\{\begin{array}{l} t_{n}\\ t_{s}\\ t_{s} \end{array}\right\}=\left[\begin{array}{ccc} K_{n} & 0 & 0\\ 0 & K_{s} & 0\\ 0 & 0 & K_{t} \end{array}\right]\left\{\begin{array}{l} \delta_{n}\\ \delta_{s}\\ \delta_{s} \end{array}\right\}$$

The cohesive element caused damage when a quadratic stress interaction function involving the nominal stress ratios reached 1. This damage criterion can be written as follows:(5)$${\left(\frac{\langle \sigma_{n}\rangle }{N_{\max}}\right)}^{2}+{\left(\frac{\langle \sigma_{t}\rangle }{T_{\max}}\right)}^{2}+{\left(\frac{\langle \sigma_{s}\rangle }{S_{\max}}\right)}^{2}=1$$
where $\sigma_{i}(i=n,s,t)$ are the normal and in-plane stresses and $N_{\max},T_{\max},S_{\max}$ are the corresponding maximun stresses in three directions.

The mix-mode fracture energy-release rate criterion of B-K [23] is particularly useful in describing the damage evolution of the cohesive element, as follows:(6)$$G_{TC}=G_{IC}+\left(G_{IIC}-G_{IC}\right){\left(\frac{G_{IIC}+G_{IIIC}}{G_{IC}+G_{IIC}+G_{IIIC}}\right)}^{\eta }$$
where $G_{IC}$, $G_{IIC}$ and $G_{IIIC}$ are the critical fracture energy-release rate in the normal, first, and second shear directions, respectively, while η is a cohesive element parameter set to 1.45.

#### 3.2.2. FEM Model

The DCB tests were simulated by using the ABAQUS/Standard, according to the specimen’s geometry and the experimental conditions, as shown in Figure 8. Two kinds of element types were used in this DCB model. The composite laminate was modeled with 26,000 three-dimensional eight-node linear brick elements (C3D8). In order to simulate the delamination, 2600 linear cohesive elements were applied in the interlamiar material.

The material properties used in this model are shown in Table 3. In particular, the critical energy-release rates used to simulate cohesive element damage initiation and the propagation of the interlaminar material were set from the experimental average value.

The finite element simulation results of the original specimen and the MWCNTs toughening specimen are shown in Figure 9. It can be seen that the deformation contour of both were similar, but the deformation of the MWCNTs toughening specimen was larger, which was consistent with the experimental results.

Figure 10 shows the peel stress distribution and interlaminar damage associated with a load displacement increase for the original specimens. As can be seen from the SDEG state of the cohesive layer, the interlayer cracks expanded first from the middle and then extended to both ends in a V-like expansion pattern.

In Table 3, the Young’s modulus *E_11_* of the composite was experimentally measured. By finite element simulation, the average *G_IC_* value of the original specimens was 1200 J/m^2^, and the average *G_IC_* value of the MWCNTs toughening specimens was 2300 J/m^2^, which was not significantly different from the experimental data processing value stated in Section 3.1. The original group matched fairly well and the error of the MWCNTs group was less than 6%. Figure 11 shows a load-displacement comparison between the experiment and the simulation of original and MWCNTs specimens. It can be seen that the experimental load-displacement curve was in good agreement with the simulation curve, indicating that the simulation results were reliable.

### 3.3. Analysis of Fracture Appearance

According to the results of the experimental and simulated load-displacement and *G_IC_* value, the interlaminar toughening strategy by the heat-repairing instrument with MWCNTs, as put forward by this paper, was successful and significantly effective in field-repair. However, only the mechanical properties were obtained from the experimental data, and the toughening mechanism is still unknown. In order to identify the interlaminar toughening mechanism, an analysis of the fracture appearance was needed. Hence, we carried out an analysis of three levels of fracture appearance: macro-scale fracture appearance photographed by an optical microscope, and meso-scale and micro-scale fracture appearance by SEM.

#### 3.3.1. Macro-Scale Analysis

Figure 12 shows the macroscopic fracture morphology photos of the original and MWCNTs toughening specimens, with optical microscope magnification of four times. The fracture surface of the original specimen (Figure 12a) appears to be smooth, and the MWCNTs toughening specimen’s surface (Figure 12b) is relatively rough. This proves that spraying MWCNTs onto the interlaminar surface of CFRPs affects the fracture appearance characteristic.

#### 3.3.2. Micro-Scale Analysis

Figure 13 is the fracture appearance of an original specimen photographed by SEM. Figure 13a shows the meso-scale fracture appearance of the fiber bundle on the composite woven belts, magnified 200 times. It can be observed that the warp bundle was vertical in relation to the weft one, and the cross-corners were filled with resin. The surfaces of the fiber bundles and the resin were smooth and lacking in the fiber-bridging phenomenon. Figure 13b,c are the micro-scale fracture appearances of the fiber bundle, magnified 1500 and 5000 times, respectively. Fiber-matrix (F-M) interface debonding and matrix cracking are clearly visible in the figures. The F-M debonding surfaces were clean and smooth, indicating that the interface was relatively weak and the debonding happened prior to matrix cracking. In the SEM photographs, fiber bundle breakage, matrix cracking, and interface debonding of the F-M can be easily observed. The matrix cracking and the F-M interface debonding were the major modes of fracture.

Figure 14 shows meso-scale and micro-scale SEM photographs of the fracture appearance of an MWCNTs toughening specimen. Figure 14a shows the meso-view of the bundle of the composite woven belts by SEM, magnified 200 times. Figure 14b–e show the fiber bundle’s micro-scale fracture appearances, magnified 1500 and 5000 times. Figure 14f shows the MWCNTs pull-out picture, magnified 100,000 times by SEM. The fiber-bundle surfaces were rough and lacking any obvious fiber-bridging phenomenon. Small quantities of fiber breakage and matrix peeling-off, the major modes of fracture, represented the large amount of F-M debonding and matrix cracking in mountain shape. The extensive matrix cracking in mountain shape, in which the MWCNTs were embedded, led to the rough fracture surface. The embedded MWCNTs were unbroken, due to the high strength. The fracture mode was the MWCNTs pull-out of the matrix, with 1 μm.

### 3.4. Summary of the Toughening Mechanism of MWCNTs

According to the macro-, meso-, and micro-scale analyses of the fracture appearance, the pre-cracks propagated along the preconceived interlaminar surface. Due to the warp and weft bundles binding to each other, there was little fiber bridging and the cracks deflected to the pure matrix of relatively low strength, which provided a great enhancement of interlaminar toughness with small dispersibility. The typical interlaminar-toughening mechanism of MWCNTs are shown in Figure 15. As shown in Figure 15a, when the crack of Model I received tensile stress, the damage sequence of the original specimens, without MWCNTs, occurred as follows: F-M interface debonding occurred first; then, the debonding grew into the adjacent matrix and proceeded to be new cracks with smooth appearances, are vertical to the tensile stress direction; finally, the F-M interface debonding cracks connected with each other until the interlaminar surface separated completely. However, with regard to the MWCNTs toughening specimens, the cracks of F-M interface debonding grew into the matrix, then deflected upon meeting the MWCNTs, and finally gave rise to the fracture appearance in mountain shape, as shown in Figure 15b. Thus, adding MWCNTs increased the crack length and enlarged the fracture surface area, which led to an increase in *G_IC_*. On the other hand, the MWCNTs pull-out resulted in the fracture toughness improving remarkably, which was the secondary major mechanism of the interlaminar toughening.

## 4. Conclusions

This paper adopted an experimental and simulation approach to study the improvement effect and the toughening mechanism of MWCNTs on the Mode I interlaminar toughness of CFRPs, using a heat-repairing instrument in applying field-repair technology. The following conclusions can be drawn:This paper proposed a convenient, low-cost, and efficient field-repair approach for enhancing the interlaminar fracture toughness of CFRPs. For Mode I fracture toughness, MWCNTs can greatly enhance interlamilar toughness, compared with PWCs. The average value of *G_IC_* increased by 102.92%. Despite removing the maximum *G_IC_* value in the MWCNTs group, it continued to show 95% improvement. The density of the MWCNTs was 1.58 g/m^2^.The load-displacement curves of the original and MWCNTs-toughening groups behaved in three stages: linear increasing, nonlinear increasing, and nonlinear decreasing. Compared with the original group, not only did the maximum load peak of MWCNTs toughening group increase by 19.5%, but the slope rate of nonlinear decreasing phase also decreased, by 63.8%. Both changes indicated that adding MWCNTs at each interface between the prepreg layers improves toughness.In comparing the simulation with the experiment, the good agreement of load-displacement and *G_IC_* value confirmed that the FEM model and the mechanical parameters were validated. The built-in cohesive element with a bilinear traction-separation relationship was utilized to capture delamination, and the mix-mode criterion of B-K described the damage evolution of the cohesive element, which was suitable for the original epoxy resin and MWCNTs interlaminar simulation.The fracture appearances of original group were smooth and without obvious fiber bridging, but the surfaces of the MWCNTs group were rough. A large amount of F-M interface debonding and matrix cracking in mountain shape were the major modes of fracture, with little fiber breakage and matrix peeling. The reason for the rough surface is found in the embedded MWCNTs. The initiated cracks of F-M interface debonding deflected in the matrix when meeting MWCNTs, which led to the fracture appearance in mountain shape. Such a fracture appearance in mountain shape caused the crack length to increase and led to *G_IC_* improvement.

## Figures and Tables

**Figure 1 materials-15-07691-f001:**
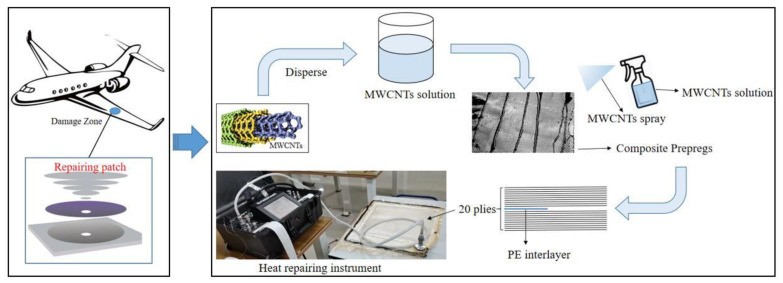
Illustration of the MWCNTs-based field-repair strategy.

**Figure 2 materials-15-07691-f002:**
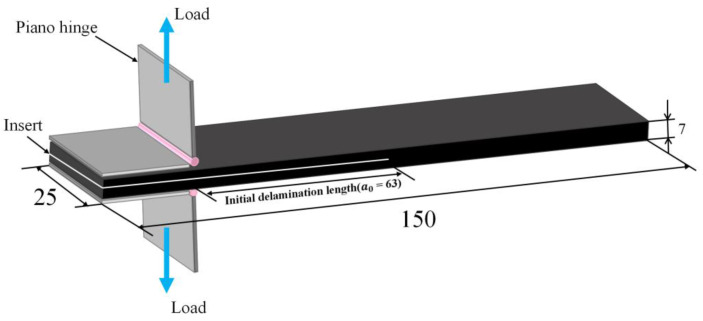
Schematic illustration of the DCB specimen and loading conditions (unit: mm).

**Figure 3 materials-15-07691-f003:**
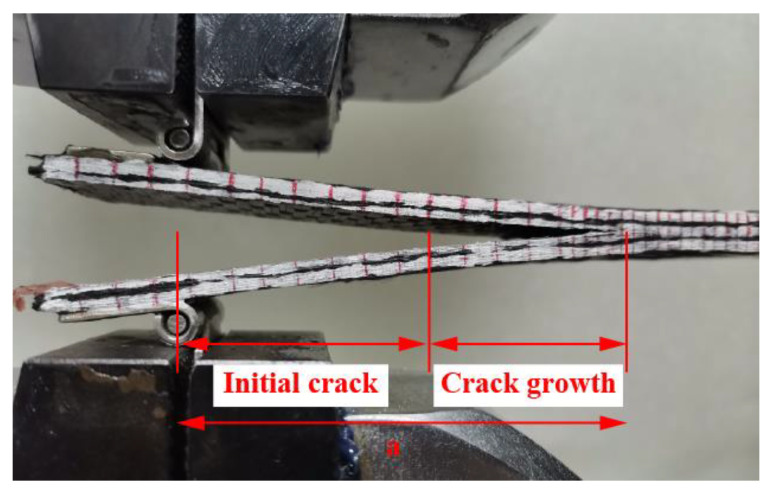
Illustration of the initial crack and the crack growth lengths in the DCB specimen.

**Figure 4 materials-15-07691-f004:**
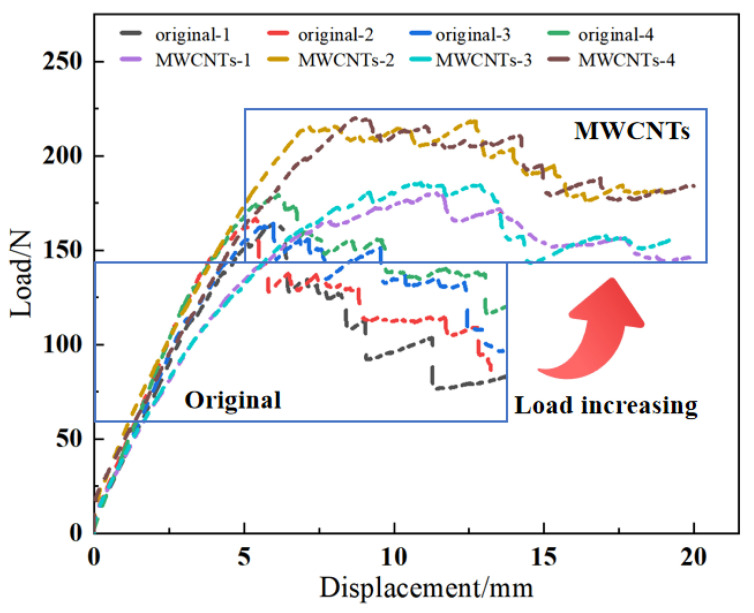
Load-displacement curves from the DCB test.

**Figure 5 materials-15-07691-f005:**
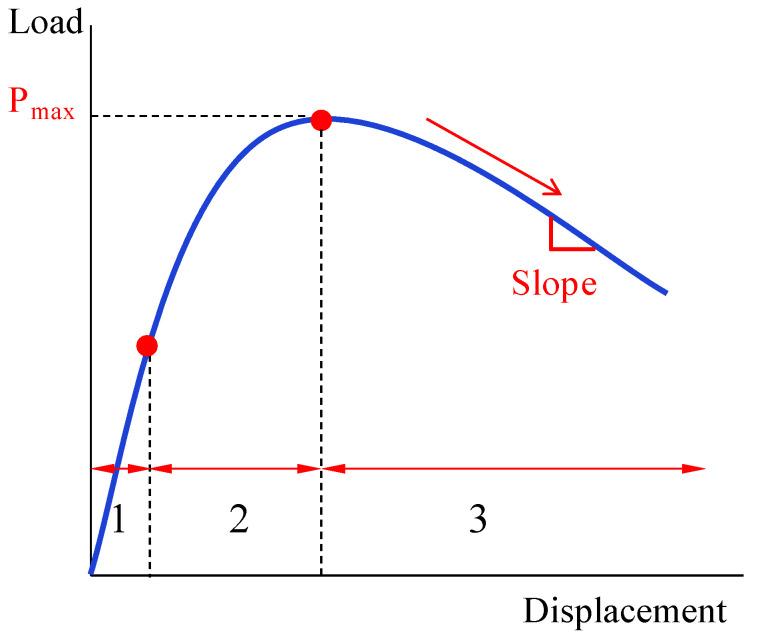
The simplified load-displacement curve for the DCB test.

**Figure 6 materials-15-07691-f006:**
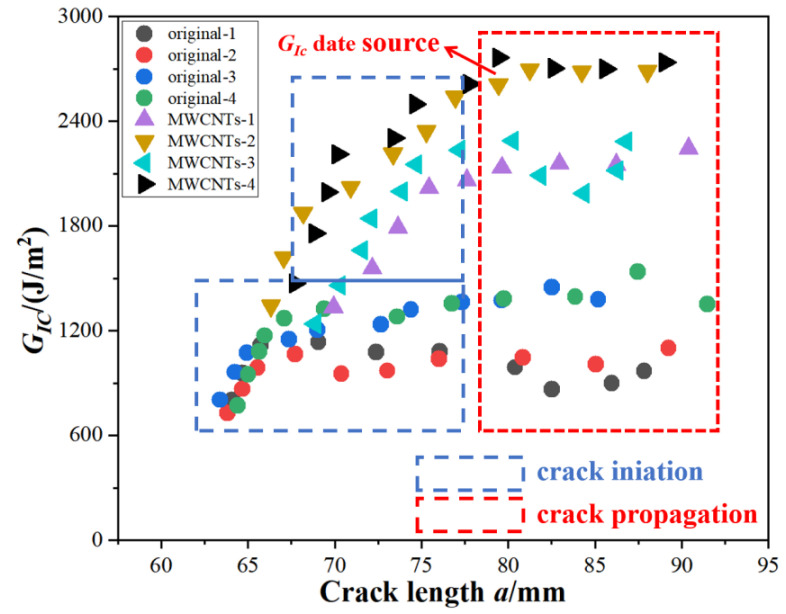
The *G_IC_-a* relational graph.

**Figure 7 materials-15-07691-f007:**
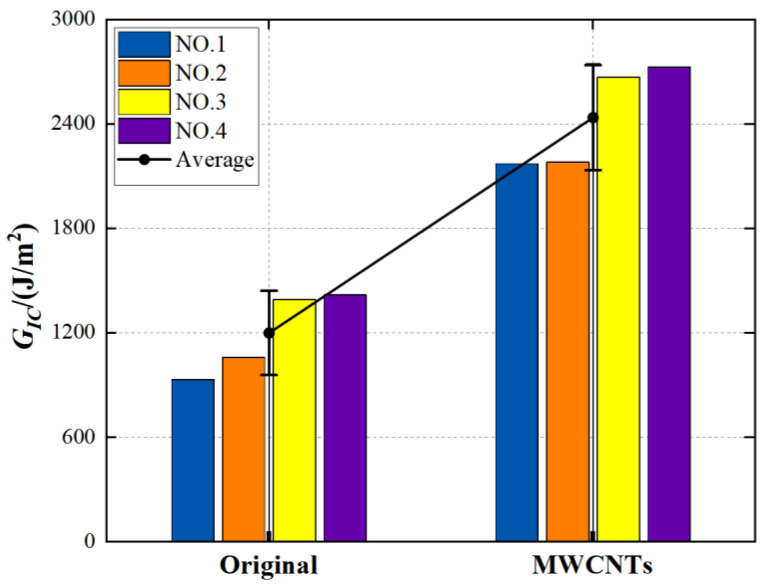
The *G_IC_* variation for the original group and the MWCNTs toughening group.

**Figure 8 materials-15-07691-f008:**
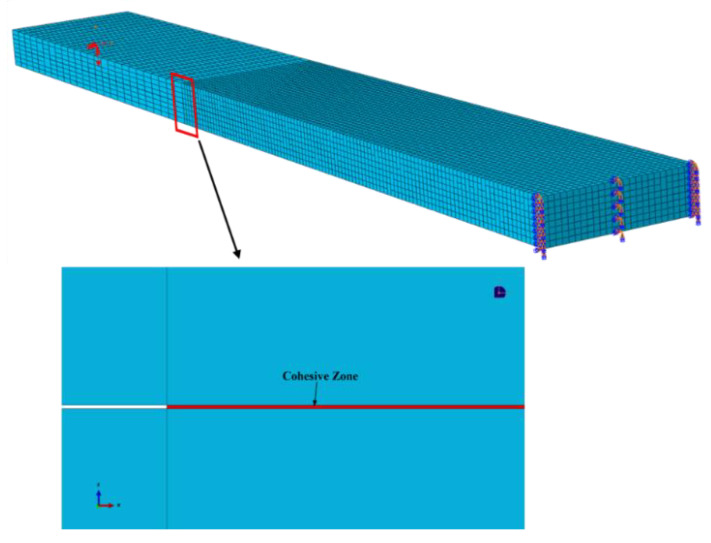
The specimen’s geometry and experiment conditions.

**Figure 9 materials-15-07691-f009:**
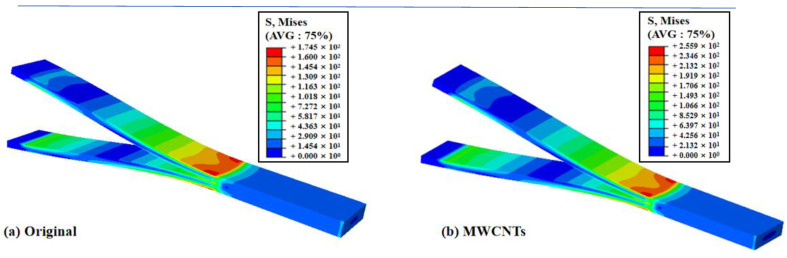
The deformation contour of the original specimens and the MWCNTs toughening specimens.

**Figure 10 materials-15-07691-f010:**
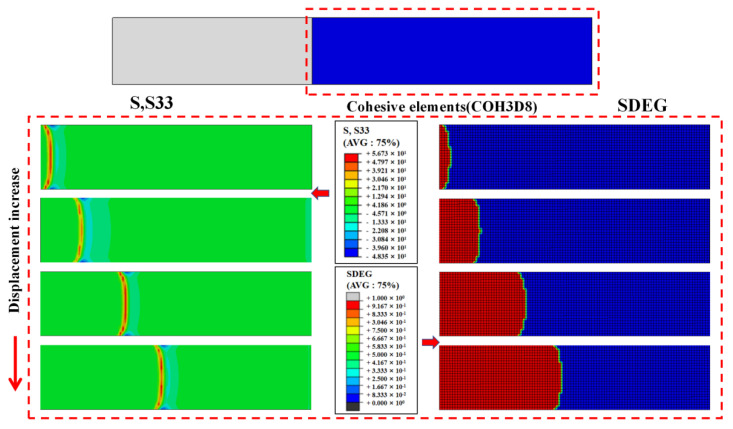
Peel stress distribution and interlaminar damage associated with the load displacement increasing for the original specimens.

**Figure 11 materials-15-07691-f011:**
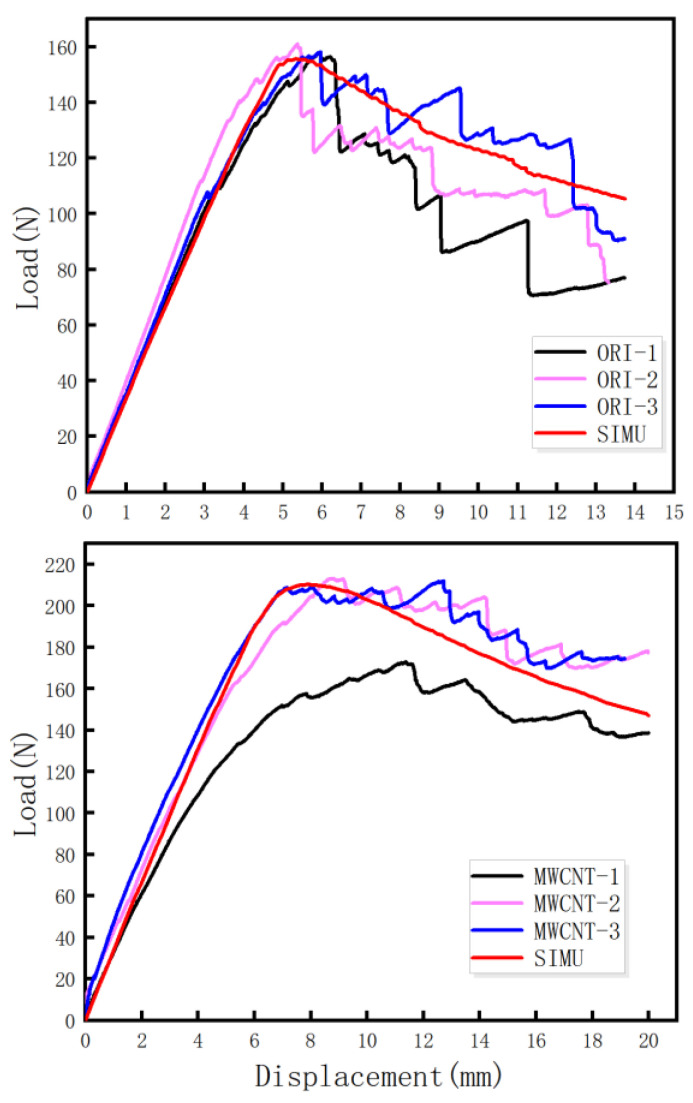
Load-displacement comparison between the experiment and the simulation of the original and MWCNTs toughening specimens.

**Figure 12 materials-15-07691-f012:**
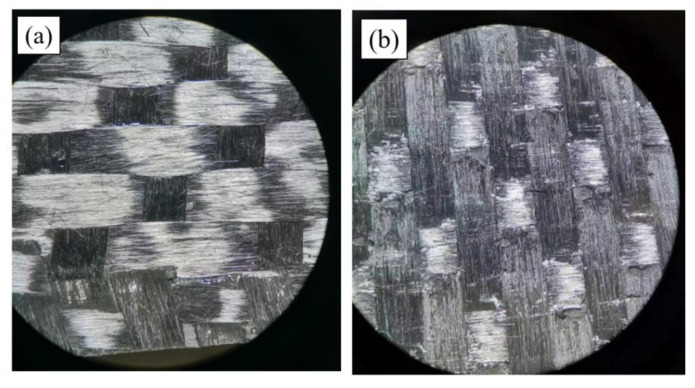
Photos by optical microscope: (**a**) original specimen; (**b**) MWCNTs toughening specimen.

**Figure 13 materials-15-07691-f013:**
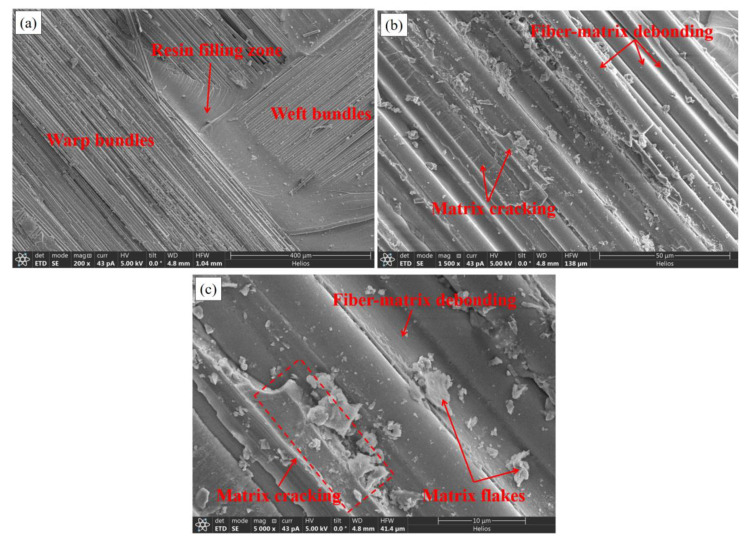
SEM photos of original specimen: (**a**) magnified 200 times; (**b**) magnified 1500 times; and (**c**) magnified 5000 times.

**Figure 14 materials-15-07691-f014:**
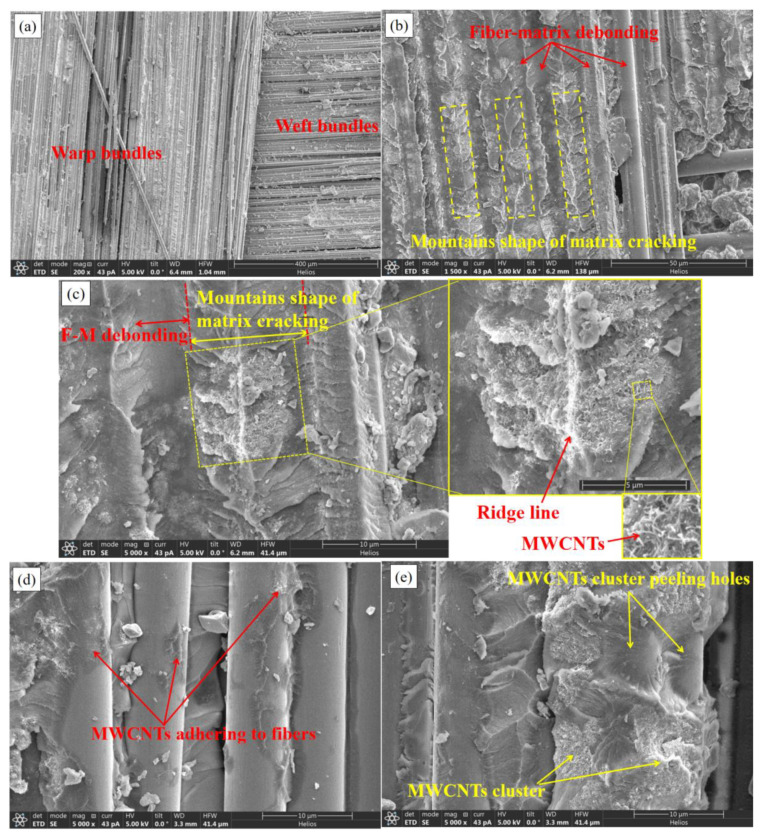
Meso- and micro-scale SEM photos of the fracture appearance of a MWCNTs toughening specimen. (**a**) magnified 200 times, (**b**) magnified 1500 times, (**c**–**e**) magnified 5000 times, (**f**) magnified 10,000 times.

**Figure 15 materials-15-07691-f015:**
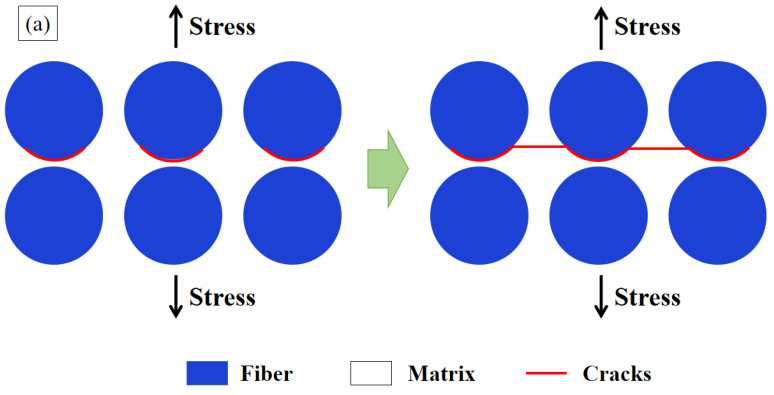
Schematic of fracture mechanism: (**a**) original specimens; (**b**) MWCNTs toughening specimens.

**Table 1 materials-15-07691-t001:** Material properties of carbon fibers and epoxy resin.

Materials	Type	Elongation to Break (%)	Tensile Modulus (GPa)	Tensile Strength (MPa)	Density (g/cm^3^)
Carbon fiber	T700-3K	2.0	240	4900	1.80
Epoxy resin	EA 9390 AERO	2.5	2.88	56.5	1.15

**Table 2 materials-15-07691-t002:** The maximum loads and nonlinear decreasing slopes of the original specimens and the MWCNTs toughening specimens.

Specimens	P_max_/N	Average Value	Increasing Rate/%	Slope/%	Average Value	Decreasing Rate/%
Original	163.2	168.3	-	10.9	10.5	-
	166.9			10.9		
	164.0			11.0		
	179.2			9.1		
MWCNTs toughening	180.5	201.2	19.5% ↑	3.9	3.8	63.8% ↓
	218.5			5.2		
	185.7			3.0		
	219.9			3.2		

**Table 3 materials-15-07691-t003:** The material properties.

Materials	Parameters	Values
Composite	Young’s modulus	$E_{11}=E_{22}=E_{33}=20\;\mathrm{GPa}$
	Shear modulus	$G_{12}=G_{13}=G_{23}=3.3\;\mathrm{GPa}$
	Poisson’s ratio	$\nu_{12}=\nu_{13}=\nu_{23}=0.32$
Cohesive	Modulus	$E_{\mathrm{nn}}=E_{ss}=E_{tt}=1.2\times 10^{6}$ N/mm^2^
	Strength	$\sigma_{n}=50\;\mathrm{MPa},\sigma_{s}=\sigma_{t}=100\;\mathrm{MPa}$
	Fracture Energy	(Original) $G_{IC}=1200\;J/m^{2},G_{IIC}=G_{IIIC}=1600$ J/m^2^
		(MWCNT) $G_{IC}=2300\;J/m^{2},G_{IIC}=G_{IIIC}=3000\;J/m^{2}$

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
