# Peer review of "A CNT-Toughened Strategy for In-Situ Repair of Aircraft Composite Structures"

_materials, 2022, doi:10.3390/ma15217691_

Round 1

Reviewer 1 Report

This paper presented aspects regarding the in-situ repair of aircraft composite structures using carbon nanotubes (CNTs). The use of CNTs to strengthen and toughen interphase of composites is a facile and efficient strategy [https://doi.org/10.1016/j.compositesb.2022.109785]. In this study, MWCNTs are utilized to enhance the interlaminar toughness of PWC laminates by a low-cost and rapid field repair strategy.

The paper is sound, original, interesting, well-organized and clearly presented.

The references are in accordance with the subject approached. Authors could take into account to add the reference [https://doi.org/10.1016/j.compositesb.2022.109785] and also to find other recent publications on the topic addressed.

Please make the following corrections:

Page 4, line 1 – “2.2. Specimens manufacturing” instead of cut one “2.1. Specimens manufacturing”

Page 6 – “3. Experimental and simulation results and analysis / 3.1. Experimental behavior, crack length a and energy release rate GIC” instead of “2. Experimental and simulation results and analysis / 2.1. Experimental behavior, crack length a and energy release rate GIC”; Be careful when numbering the chapters and subchapters!

Page 9 – “as shown in Fig. 8” instead of “as shwon in Fig. 8”

Page 12 – “It can be observed that warp…” instead of “It can be obeserved that warp…”

Author Response

Thank you for giving us the opportunity to submit a revised draft of the manuscript “A CNT toughened strategy used for in-situ repair of aircraft composite structure” for publication in the Journal of material. We appreciate the time and effort that you and the reviewers dedicated to providing feedback on our manuscript and are grateful for the insightful comments on and valuable improvements to our paper. We have incorporated most of the suggestions made by the reviewers.

We feel great thanks for your professional review work on our article. As you are concerned, there are several problems that need to be addressed. According to your nice suggestions, we have made extensive corrections to our previous draft, the detailed corrections are listed below.

Page 4, line 1 – “2.2. Specimens manufacturing” instead of cut one “2.1. Specimens manufacturing”

Page 6 – “3. Experimental and simulation results and analysis / 3.1. Experimental behavior, crack length a and energy release rate GIC” instead of “2. Experimental and simulation results and analysis / 2.1. Experimental behavior, crack length a and energy release rate GIC”; Be careful when numbering the chapters and subchapters!

Page 9 – “as shown in Fig. 8” instead of “as shwon in Fig. 8” 

Page 12 – “It can be observed that warp…” instead of “It can be obeserved that warp…”

Thank you for your reminder. We have corrected the above four problems in the manuscript.

Reviewer 2 Report

The manuscript is interesting and well written, and it can be published after a minor revisor. I have one minor comment other than the manuscript is acceptable.

1. Provide scale for Fig 12.

Author Response

Thank you for giving us the opportunity to submit a revised draft of the manuscript “A CNT toughened strategy used for in-situ repair of aircraft composite structure” for publication in the Journal of material. We appreciate the time and effort that you and the reviewers dedicated to providing feedback on our manuscript and are grateful for the insightful comments on and valuable improvements to our paper. We have incorporated most of the suggestions made by the reviewers.

We feel great thanks for your professional review work on our article. As you are concerned, Figure 12 shows the macroscopic fracture morphology photos of the Original and MWCNTs toughening specimen with optical microscope magnification of 4 times. 

Reviewer 3 Report

The manuscript deals with the development and evaluation of a procedure for in-situ repair of composites used in aircrafts. Some of the mechanical properties of the composites were studied experimentally and by simulation. The topic is worth to study. Sadly, the experiments are poorly presented. The results and discussions could be improved and presented in a clearer manner. Various revisions are required.

·         Text editing is required e.g. in many cases there are more than two spaces between two words. The numbering of sections is wrong.

·         English improvement is required throughout the text, e.g. in the abstract “... is founded…” should be “was found”. In page 2 similarly “…founded by Lijima…”.

·         Page 2, first line. “…while sacrificing the in-plane mechanical properties”. Please provide more information for this.

·         Page 3 . “….However, the field repair process only requires 1 bar of curing pressure by a vacuum bag…….” . If 1 bar of pressure is required then why vacuum is needed?

·         Polyethylene and its characteristics are not mentioned in Section 2.1.

·         Section “2.4. Double cantilever beam (DCB) set-up and tests”. Equations 1 and 2. How Iz is calculated?

·         The purpose of the Experimental Part Section is to provide adequate information if one wants to repeat the experiment. By reading the experimental part I don’t think I will be able to repeat the experiment and prepare the same material. Figure 1 is suitable for graphical abstract, however, another figure should be prepared to describe step by step the overall process. Much more experimental details should be added in the text as well.

·         Section “2. Experimental and simulation results and analysis” should have “3” as numbering and not “2”.

·         Section simulation Method and Results instead of 2.1 should be 3.2. Other subsections also have wrong numbering.

·         Numbering is in wrong position in equation 3.

·         The values presented in Table 3, were derived by experiments, literature or both?

·         Table 3. What is the difference betweenYoung’s modulus and Modulus?

·         Table 3. The units of Modulus are N/mm and not N/mm2?

·         Table 3. G is usually mentioned as shear modulus and not Young’s modulus.

·         Table 3. Poisson’s ratio is written incorrectly (Possion’s)

·         Figure 12. If possible, please provide figures from both samples under the same magnification.

·         Figure 13 and section 3.3.2. What do you mean by matrix? The epoxy resin is the matrix?

Author Response

Thank you for giving us the opportunity to submit a revised draft of the manuscript “A CNT toughened strategy used for in-situ repair of aircraft composite structure” for publication in the Journal of material. We appreciate the time and effort that you and the reviewers dedicated to providing feedback on our manuscript and are grateful for the insightful comments on and valuable improvements to our paper. We have incorporated most of the suggestions made by the reviewers.

We feel great thanks for your professional review work on our article. As you are concerned, there are several problems that need to be addressed. According to your nice suggestions, we have made extensive corrections to our previous draft, the detailed corrections are listed below.

  • Page 2, first line. “…while sacrificing the in-plane mechanical properties”. Please provide more information for this.
  • Shin Y C, Kim S M. Enhancement of the interlaminar fracture toughness of a carbon-fiber-reinforced polymer using interleaved carbon nanotube buckypaper. Applied Sciences, 2021, 11(15): 6821.

This paper says that the interleaving the film of a laminate-type composite poses the risk of deteriorating the in-plane mechanical properties.

  • Page 3 . “….However, the field repair process only requires 1 bar of curing pressure by a vacuum bag…….” . If 1 bar of pressure is required then why vacuum is needed?

Because the vacuum bag is used for vacuum curing.

  • Polyethylene and its characteristics are not mentioned in Section 2.1.

Polyethylene (PE) films with a thickness of 0.03 mm be inserted at the midplane of the laminate during layup to form an initiation site for the delamination, The total length is about 75 mm.

  • Section “2.4. Double cantilever beam (DCB) set-up and tests”. Equations 1 and 2. How Iz is calculated?

Iz is the moment of inertia of one arm, b is the width of the specimen, h is half the thickness of the specimen.

  • The purpose of the Experimental Part Section is to provide adequate information if one wants to repeat the experiment. By reading the experimental part I don’t think I will be able to repeat the experiment and prepare the same material. Figure 1 is suitable for graphical abstract, however, another figure should be prepared to describe step by step the overall process. Much more experimental details should be added in the text as well.

Fig. 1 Illustration of the MWCNTs-based field repair strategy

The MWCNTs-based field repair technological process of composite specimens is shown in Fig. 1. The first step is dispersing the MWCNTs into the acetone solution under ultrasonic wave for 15 mins at 23℃ (ambient temperature) to avoid the agglomeration of MWCNTs, and spraying the nano-materials solution on the each surface of prepregs by sprayer. The composite prepregs are made by coating the epoxy resin uniformly on the carbon fiber cloth. And the supplements density of 1.58 g/m2 is chosen to investigate and obtain the interlaminar toughness enhancement. And then, 20 wet composite prepregs are totally stacked with the predefined sequence, in which the 0° and 90° plies are alternately laid. Meanwhile, polyethylene (PE) films are inserted between the 10-th and 11-th layers, to represent the initial Mode I cracks, namely the LVI-induced delamination. After that, the wet fabrics are cured by a heat repairing instrument with the curing temperature of 120℃ and the pressure of 1 bar for 4 h.

  • Text editing is required e.g. in many cases there are more than two spaces between two words. The numbering of sections is wrong.
  • English improvement is required throughout the text, e.g. in the abstract “... is founded…” should be “was found”. In page 2 similarly “…founded by Lijima…”.
  • Section “2. Experimental and simulation results and analysis” should have “3” as numbering and not “2”.
  • Section simulation Method and Results instead of 2.1 should be 3.2. Other subsections also have wrong numbering.
  • Numbering is in wrong position in equation 3.
  • Table 3. G is usually mentioned as shear modulus and not Young’s modulus.
  • Table 3. Poisson’s ratio is written incorrectly (Possion’s)

We were really sorry for our careless mistakes. Thank you for your reminder. We have corrected the above problems in the manuscript.

  • The values presented in Table 3, were derived by experiments, literature or both?

The values presented in Table 3, were derived by experiments and literature.

  • Table 3. What is the difference betweenYoung’s modulus and Modulus?

During the modeling process, the parameter data of the carbon fiber layer and the cohesive layer are different.

         Table 3. The units of Modulus are N/mm and not N/mm2?  

The units of Modulus are .

  • Figure 12. If possible, please provide figures from both samples under the same magnification.

Figure 12 shows the macroscopic fracture morphology photos of the Original and MWCNTs toughening specimen with optical microscope magnification of 4 times.

  • Figure 13 and section 3.3.2. What do you mean by matrix? The epoxy resin is the matrix?

Yes,the matrix is the epoxy resin.
